# Facing Societal Challenges in Living Labs: Towards a Conceptual Framework to Facilitate Transdisciplinary Collaborations

**Indre Kalinauskaite** [1,*] 📧, **Rens Brankaert** [1,2] 📧, **Yuan Lu** [1], **Tilde Bekker** [1], **Aarnout Brombacher** [1] **and Steven Vos** [1,3] 📧

1. Systemic Change Group, Department of Industrial Design, Eindhoven University of Technology, 5612AP Eindhoven, The Netherlands; r.g.a.brankaert@tue.nl (R.B.); Y.Lu@tue.nl (Y.L.); m.m.bekker@tue.nl (T.B.); a.c.brombacher@tue.nl (A.B.); s.vos@tue.nl (S.V.)
2. School of Allied Health Professions, Fontys University of Applied Sciences, 5612MA Eindhoven, The Netherlands
3. School of Sport Studies, Fontys University of Applied Sciences, 5600AH Eindhoven, The Netherlands
* Correspondence: i.kalinauskaite@tue.nl

**Abstract:** Living labs are an extremely attractive open innovation landscape for collaborative research and development activities targeting the complexity of today's societal challenges. However, although there is plenty of support for collaboration, we still lack clear guidelines to direct transdisciplinary stakeholder networks of academics and practitioners through collaboration processes in the living lab ecosystem. In other words, we lack answers to the question of "how to collaborate?" In the present paper we propose a conceptual framework defining relevant stages to initiate and facilitate transdisciplinary collaboration processes. We base our framework on collaboration challenges described in the literature, specifically the need for stakeholder alignment, as well as challenges experienced in practice, which we report through exploratory case studies. In the proposed conceptual framework, we advocate the application of co-creation methods, both at the level of the living lab (macro) and in projects (meso) within the living lab, in order to define, with all involved parties and stakeholders, the scope and strategy of the living lab and to facilitate stakeholder alignment. Additionally, we integrate an iterative approach and a feedback loop in order to account for the dynamic nature of the collaboration process and to enable reflection and evaluation.

**Keywords:** transdisciplinary collaboration; co-creation; living lab

## 1. Introduction

Currently, the world is facing a myriad of complex socio-technical challenges, from irreversible climate change to the immense societal impact resulting from a global pandemic, such as COVID-19, or widespread physical inactivity. Perhaps the most distinguished global strategy to tackle these challenges in a collaborative manner is reflected in the Sustainable Development Goals of the United Nations [1]. A collection of 17 interlinked goals represent a blueprint towards a sustainable and prosperous future for all. Here, scientists, governments, industries, and citizens are invited to join their effort to successfully combat the world's problems.

A living lab, whether regarded as an approach [2], an ecosystem [3], or a milieu for innovation [4,5], offers a variety of benefits for research and development activities targeting societal challenges. Although, there is no consensus on the definition of the living lab to date [6], multi-stakeholder collaboration and end user involvement are the core elements of a living lab approach [7–13]. As such, multi-stakeholder collaboration in living lab projects is considered a requirement for innovation [6] and is often brought forward using innovation frameworks, such as triple-, quadruple-, or quintuple-helix models (e.g., [2]). Today, we see a myriad of successful examples implementing the living lab approach to directly or indirectly tackle specific societal challenges through collaboration, for example

to stimulate the transition towards a carbon neutral future [14], to deal with cultural diversity and social cohesion issues [15], or to fast forward the innovation in healthcare [16]. Most of the corresponding literature describes the added value of collaboration for innovation, in particular, how collaboration between industry, academic, and governmental sectors promotes knowledge-based economic growth and social development [17–21]. The user and citizen involvement in the collaboration process is often discussed through co-creation, which enables user/citizen participation in the development of necessary and useful products and services [5]. Meanwhile, the collaboration processes between living lab stakeholders is mainly considered in the context of the organization of the living lab. For example, the nature of collaboration may be used to determine a typology of living lab (e.g., enabler vs. user driven living labs, [20]), while the extent of diversity in stakeholder network may determine the overall success of collaborative innovation processes [21]. In the context of societal challenges, precisely the collaboration processes in the living labs have been shown to significantly contribute to the development of sustainable public policy and in turn result in greater social impact [22]. However, Greve et al. [23] suggest that the applied nature of the research activities in the living labs has amounted to extensive empirical evidence and currently calls for a more theoretical approach; also, towards topics such as collaboration processes [10,24].

Collaboration is vital for the living lab [6] and challenges in collaboration processes might stagnate or even terminate the developments in the living lab [25]. Interestingly, however, the dynamic nature of collaboration processes and the challenges thereof seem to be overlooked in the living lab literature [26]. A lack of theoretical and practical guidelines on how to collaborate—initiate, facilitate, maintain, and evaluate the collaboration process between heterogenous groups of stakeholders—poses serious challenges in advancing living lab research. In the present manuscript, we explore the theoretical and practical challenges of transdisciplinary collaboration through a narrative literature review and three exploratory case studies. The main objective of this paper is to develop a conceptual framework to help practitioners and researchers effectively set up and facilitate transdisciplinary collaborations in complex multi-stakeholder socio-ecological systems, such as living labs. In particular, we focus on the early stages of collaborative initiatives following the decision to collaborate. In the proposed conceptual framework, we stress the important role of co-creation—a collaborative definition of the problem, as well as co-design and co-implementation of the solution—as a means to achieve and maintain alignment amongst heterogeneous group of collaborating parties (i.e., stakeholders) in different stages of collaboration. Additionally, we argue that implementing co-creation on meso (project) and macro (living lab) layers of the living lab ecosystem will benefit both the collaboration processes on strategic and implementation levels and our understanding of how different aspects of the living lab ecosystem interrelate. The conceptual framework presented in this paper was designed based on collaboration challenges reported in the literature and lessons learned in practice.

The structure of this paper is as follows. We first present the literature exploring the nature and challenges of transdisciplinary collaboration. Then, we discuss the potential role of co-creation to enhance stakeholder alignment and to facilitate the transdisciplinary collaboration in the living lab setting. Next, we introduce the exploratory case studies and the results thereof, followed by the introduction of a conceptual framework to set up and facilitate transdisciplinary collaboration in a complex socio-ecological context, such as in living labs. We conclude this paper with a discussion of the implications of our contribution to living lab theory and practice.

## 2. Theoretical Background

### 2.1. Nature and Challenges of Transdisciplinary Collaboration

Transdisciplinary collaboration enables the integration of different viewpoints, methodologies, and approaches in stride with global challenges [24,27,28]. Due to its integrative nature and the fact that it is an iterative process [24], transdisciplinary collaboration offers

a huge potential to create long-term, sustainable alliances to rapidly respond to dynamic and complex nature of societal challenges [27,29], such as improving the lifelong health and well-being of all European citizens.

Transdisciplinary collaboration between a variety of disciplines and a multitude of sectors, however, is not straightforward [30,31]. In fact, Noris and colleagues [32] define transdisciplinary collaboration as a "wicked problem"; collaboration is a very complex process dependent on a spectrum of dynamic variables ranging from interpersonal relationships to complex contextual influences [33,34]. Additionally, collaborating parties in an academic context might conflict in their research methodologies and/or disagree on individual research frameworks [24,28]. In the context of living labs, stakeholders often form a very heterogeneous group of professionals encompassing, amongst other things, different disciplines, sectors, roles, and temporal dimensions [35]. The challenge here is how to facilitate the collaboration process based on stakeholders' knowledge, expertise, and/or contribution to the project development, rather than a role or function-driven, hierarchical status [36]. To overcome collaboration challenges, literature examining collaboration processes acknowledges a variety of essential conditions for successful collaboration, which often starts with creation and adoption of a shared vision [33] in the early stages of collaboration [27,37].

To ensure that a common vision is indeed shared by all, stakeholder alignment on various levels is crucial [38,39]. As such, scholars outline the importance of integration of common concepts (e.g., [34,40]) and knowledge [41], creation of shared strategy and values [42], as well as balance of mutual and individual gains [33]. The success of overall stakeholder alignment is, however, strongly dependent on how the intrapersonal, interpersonal, organizational, technological, physical environment, and political factors (e.g., as individual attitudes and values, levels of mutual respect, differences in inter-organizational culture, affinity with technology, proximity between collaborating parties, and policy) are considered throughout the design, management, and implementation of the collaboration strategy [27]. From a theoretical point of view, literature outlines ample conditions and ingredients required for successful collaboration (i.e., achieving stakeholder alignment); however, we still lack comprehensive frameworks and practical guidelines on how to operationalize this theory, as well as how to evaluate the impact and added value of transdisciplinary collaborations in practice [10,24] in living labs. In this paper, we argue that co-creation principles in multi-stakeholder ecosystems, such as living labs, have a potential to fill this gap in theory.

### 2.2. Co-Creation in Transdisciplinary Collaboration

A co-creation process involving multiple stakeholders has been defined as "a deliberate process that builds upon collaborative behaviors and attitudes, guided by a common purpose, where actors can reciprocally benefit from the co-creation outcomes" [43], p. 4039. Indeed, given that appropriate tools are provided, co-creation, by definition, facilitates inclusive creative processes, where involved parties develop shared ideas, concepts, and solutions to tackle the problems in question [44]. Co-creation, thus, scopes both co-design (collaborative definition of a problem and solution) and co-production (collaborative implementation of the solution) [45]. Evidence suggests that participation in co-creation activities, especially in the public domain, strengthens the feeling of ownership and empowerment [46]. In an open innovation context, such as living labs, co-creation is encouraged as it has been shown to foster innovation through, amongst other things, (common) value creation [47,48] and development of solutions which more effectively respond to societal challenges [45,49].

Even though co-creation is essential to user-centered and bottom up innovation driven living labs [50], some researchers suggest that it is not clear how co-creation, participation, and collaboration are organized in living labs [51,52], nor what exact tangible value it brings to participating partners [53]. Schuurman and De Marez [54] place co-creation in the living lab on a micro level, meaning it is primarily focused on involving citizens and

the end users in research and development processes [11,55], while other scholars use term of "co-creation" as a synonym for collaboration between various actors in the living lab [56]. Interestingly, with some exceptions (e.g., [57]), to our knowledge, co-creation principles are rarely structurally implemented in shaping and facilitating transdisciplinary collaboration processes in the living labs or living lab alike settings. However, in line with Pera and colleagues' [43] definition of co-creation and due to its inclusive nature, we believe that co-creation has a huge potential to facilitate stakeholder alignment in the context of transdisciplinary collaboration.

## 3. Collaboration Challenges in Practice—Exploratory Case Study

### 3.1. Materials and Methods

The focus of our exploration was on collaborative initiatives tackling the societal challenges concerning society's health and vitality. To obtain input from practice for our conceptual framework, we adopted an exploratory case study approach [58]. We explored three real-life transdisciplinary collaboration initiatives. The three cases, all in the Netherlands, were selected based on two criteria. The first criterion was the collaborative initiative's compliance with HORIZON2020's societal challenge: health, demographic change, and well-being. The specific objective of this challenge is to improve lifelong health and well-being of all European citizens. Second, the collaborating initiatives had to either employ or have an intention to employ a living lab approach and methodology in their future research and/or development activities. The cases were researched applying a critical reflection [59] method to reflect on our involvement in each case study. The anonymized descriptions of each case, stakeholder composition, and our involvement as researchers are presented in Table 1. Figure 1 presents impressions from the co-creation workshop described in Case B.

**Table 1.** Case study descriptions.

| | Case A:<br>Health and Active Lifestyle Park | Case B:<br>Healthy University Campus | Case C:<br>Cluster of Dutch Universities for Digitalization of Healthcare |
|---|---|---|---|
| **Description and aim:** | Case A is a multi-stakeholder innovation project kickstarted to develop a sports and experience park—a living lab—to support the adoption of a healthy lifestyle and social cohesion in the population in and around this park. It is an innovation project initiated and directed by the municipality of a mid-sized Dutch town in the south of the Netherlands. | Case B is a bottom up initiative steered and organized by the multidisciplinary team established at one of the Dutch universities. The aim of this initiative is to stimulate and facilitate the campus-wide adoption of a healthy lifestyle in the university population, in- and outside of the work/study context. The living lab approach is seen as a means to implement and evaluate various interventions on this university campus. | Case C is a cluster of representatives of several Dutch universities stemming from a larger national academic network (in the Netherlands) established by the association of universities in the Netherlands. The academic cluster representing case C was brought together to promote and conduct responsible digitalization in healthcare.<br>The living lab approach is broadly considered in various research and intervention activities within this cluster. |
| **Stakeholder composition:** | Multiple stakeholders from variety of sectors—industry, governance, education, non-profit organizations, foundations, and research and academic institutions. | Multiple stakeholders representing university's community—academic and non-academic staff (e.g., housing and real estate manager, community manager, sports coaches) and students. | Multiple stakeholders representing eight Dutch universities—senior and junior academic staff. |

**Table 1.** *Cont.*

| | Case A:<br>Health and Active Lifestyle Park | Case B:<br>Healthy University Campus | Case C:<br>Cluster of Dutch Universities for<br>Digitalization of Healthcare |
|---|---|---|---|
| **Our involvement:** | Our university was invited to join the living lab stakeholder network as academic partner to support the initiative in conducting transdisciplinary research and evaluating the societal impact of the initiative.<br>The first author of this paper conducted a set of semi-structured interviews with involved partners to explore and map the project's stakeholder network and to understand the underlying common vision within this stakeholder network. | The first author of this paper was asked to consult the project team on strategy development towards achieving common vision. We designed and facilitated a co-creation workshop to develop a strategic impact map for this initiative | Our university is a member of this cluster. In 2019, the first author of this paper was asked to facilitate strategic roadmap development process for this cluster. She designed and facilitated a co-creation workshop to develop a strategic impact map for this initiative. |

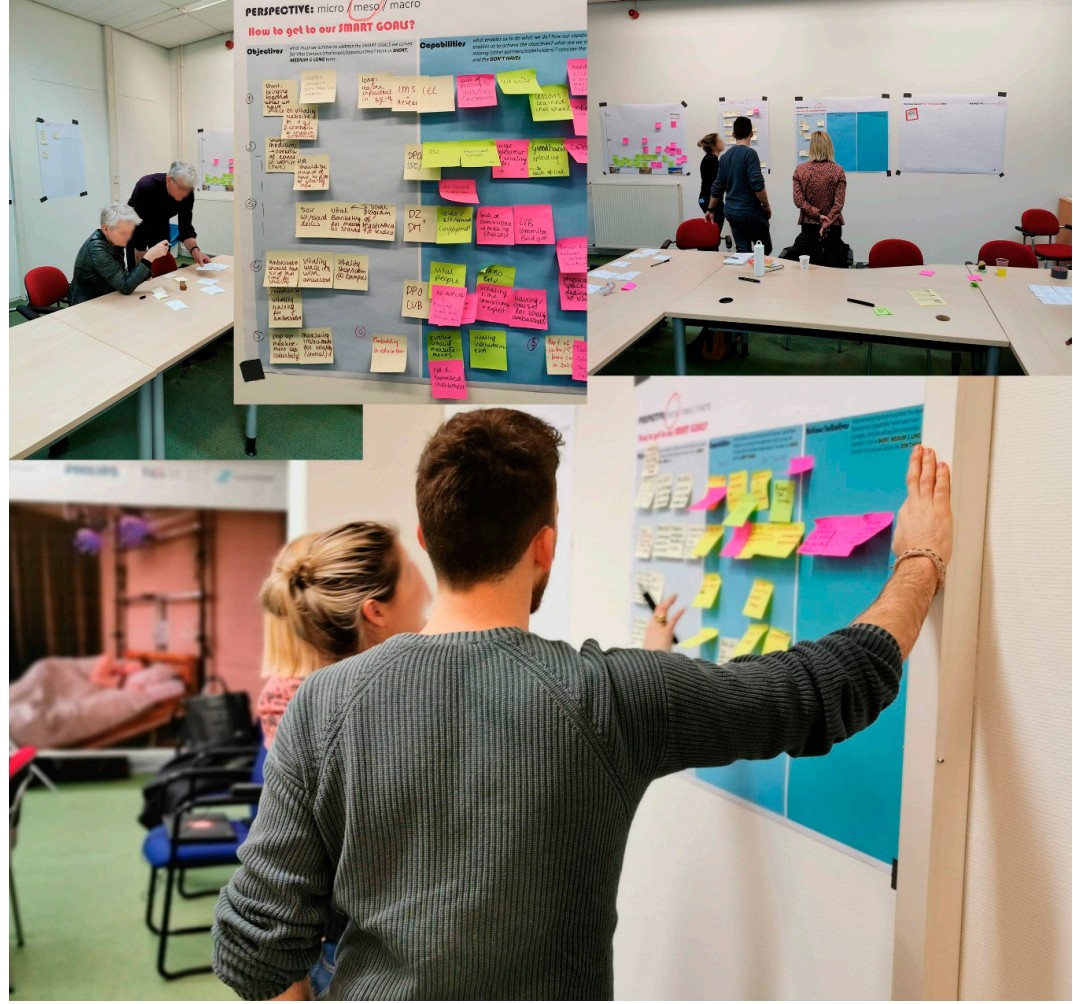

**Figure 1.** Case B impact mapping workshop.

*3.2. Results*

In this section, we briefly explore the three cases introduced in the previous section to illustrate the challenges that were encountered in practice and which might seriously hinder the collaboration process and/or success of the transdisciplinary collaborative initiatives. Insights gained through these experiences contributed to shaping the conceptual framework for approaching collaboration in the multi-stakeholder ecosystems, such as living labs.

3.2.1. Case A: Health and Active Lifestyle Park

We employed a stakeholder safari method [60] to explore and map the project's stakeholder network and to understand the underlying extent of the common vision within this stakeholder network. By interviewing core stakeholders in the project and by synthesizing information across interviews using thematic analyses [61], we learned that this project's stakeholder network is formed by a very heterogeneous group of representatives from various sectors, such as industry, government, education and academia, and non-profit. The individual stakeholders and stakeholder organizations envision themselves playing various roles in this living lab, such as technology provider, enabler, and researcher. Naturally, amongst stakeholders we detected a spectrum of different ambitions and interests, as well as variety of reasons for collaboration and involvement. Some focused on maintaining societal well-being, while the others' goal was to test their products in the field and speed up market launch. Interestingly, although all interviewed stakeholders unanimously acknowledged that everyone's involvement is crucial for the project's success, the project was owned by everyone, and at the same time by no-one, meaning there was no consensus amongst interviewed stakeholders as to who was responsible for the facilitation and maintenance of project development. Not surprisingly, there was no shared understanding about the stakeholder network structure (i.e., who else and why was involved in this project), and we noticed a lack of temporal considerations in the setup of this collaboration, especially considering the dynamics of stakeholder involvement. Finally, we sensed a lack of cohesion and were not able to distil a clear common vision. This contributed to our conclusion that in this living lab project, collaboration was treated more as a permanent state, rather than a continuous and dynamic process.

The stakeholder safari suggested the lack of alignment and overview within the stakeholder network, lack of central organization, process management, and governance, and that official project coordinators treated the living lab project and stakeholder collaboration as a straightforward and linear process.

3.2.2. Case B: Healthy University Campus

In the preparatory meetings, we learned that the project team has a clear vision to make the university's campus the most vital campus in the world by the year 2030. Although this vision was shared amongst the project team members, a far-reaching temporal span of 10 years and a very high ambition—to become "the most vital campus in the world"—made common vision rather intangible, ambiguous, and difficult to translate into actionable strategic action points. Therefore, to create a strategic impact map, a common vision had to be translated into a number of well-defined and measurable common goals—SMART goals (specific, measurable, achievable, realistic, time-based goals). In the first part of our co-creation workshop, thus, we facilitated the generation of SMART goals, stemming from a common vision. These goals were later used as a starting point to generate a strategic impact map reflecting networks capabilities and providing an overview of five strategic areas (lobbying and networking, vitality research, vitality education, vitality recourses, and vitality awareness) and core activity lines in each area (e.g., lobbying with university's executive board; providing digital vitality support).

In conclusion, this bottom-up initiative initially started with various activities and actions organized by the project team. However, as the project grew bigger, the successful interventions achieved more attention and the project team needed to adapt their gover-

nance by structuring the collaboration process, by deriving a clear long- and short-term strategy for the future, and by determining how to evaluate the progress and impact of this collaboration.

### 3.2.3. Case C: Cluster of Dutch Universities for Digitalization of Healthcare

As we started drafting the workshop for the co-creation of strategic roadmap, through discussions with involved researchers we quickly learned that several core ingredients and conditions necessary to generate the roadmap were missing. For example, the network seemed to not yet have a common vision, shared actionable goals, nor the strategy for how they intend to realize the common vision (i.e., strategic impact map). Additionally, the multidisciplinary nature of this network was treated more as an inherent transdisciplinary advantage of the network, while the potential issues that these differences might cause in achieving stakeholder alignment, such as potential disagreement between different research approaches and methodologies to tackle common research objectives, were not addressed. Therefore, we hosted a co-design workshop to guide the network members in generating a common vision, subsequent SMART goals stemming from the common vision, and finally a strategic impact map. The strategic impact map describes six strategic areas (resources; networking and lobbying; privacy and ethics; bridging science, citizens, and industry; knowledge dissemination and education; and standardization) and activities in each of these areas (e.g., writing of grant proposals, health and well-being initiatives within community). This document helped the network to clarify the strategy towards implementing a common vision and to divide the tasks and projects based on individual and group interests. The outcomes of this workshop served as input for generating a roadmap towards implementation of this cluster's goals and vision in the next stage of collaboration.

In conclusion, this network was established as a part of a top-down initiative to mobilize Dutch academic institutions. Although academic professionals were aware of the aims of the hosting organization (i.e., to promote and conduct responsible digitalization in healthcare), our exploration suggested that this network lacked a clear actionable common vision, balance between common and individual gains amongst collaborating parties, good overview of network's capabilities, and transparency about the differences in interpersonal, inter-organizational and other factors relevant for collaborative success.

### 3.2.4. Lessons Learned

In summary, every project, living lab, stakeholder, and knowledge network is different and will face different challenges depending on how the collaboration process is defined and managed [29]. In the three example cases above, we outlined different and at the same time overlaying challenges that a collaborative initiative might face, especially, in the initiation phase of a new project. Drawing from these example cases, it seems that in all of the initiatives, collaboration is seen more or less as a permanent state and not as a continuous process that can grow, change, and learn, as literature suggests [24,33]. The insights gained through our case studies confirm that guiding frameworks and implementation tools are necessary to help practitioners and academics grasp the dynamics of collaboration processes in practice, as well as to effectively govern it, e.g., [10,30]. The main misconceptions in the above-outlined examples were that collaboration was already well on its way because various project activities and sub-projects had been already started or were planned to start. However, in reality, since there was no clearly documented strategy, the collaboration from a process point of view was still in the initiation phase. As such, we conclude that too little attention was paid at structuring, organizing, and facilitating collaboration from initiation through execution to evaluation phases.

Our experience also suggests that employing co-creation methods, such as workshops employed in cases B and C, not only nurture alignment between collaborating parties—a crucial condition for successful collaboration [38,39]—but also provide invaluable "input for output". In other words, content generated during the well-designed and struc-

tured co-creation workshops is processed into invaluable tools, such as vision statements, SMART goals, or impact maps, to envision and to support the collaboration process.

## 4. Proposed Conceptual Framework

Based on the existing literature and the exploratory case study reported above, we propose a conceptual framework (see Figure 2) to successfully set up and initiate collaboration in a complex research and innovation ecosystems, such as living labs. On the one hand, our framework is focused on the initiation phase of collaborative initiatives in the context of living labs. On the other hand, the proposed integration of process dynamics in the collaboration process offers a potential to bridge the early and the advanced stages of collaboration through an evaluation process. To be more specific, we argue that the iterative nature of and the feedback loop in the proposed framework, as well as the proposed deliverables in each stage (i.e., stakeholder map, common vision, SMART goals, impact map, and roadmap), enable tracking, monitoring, and evaluation of the collaboration process.

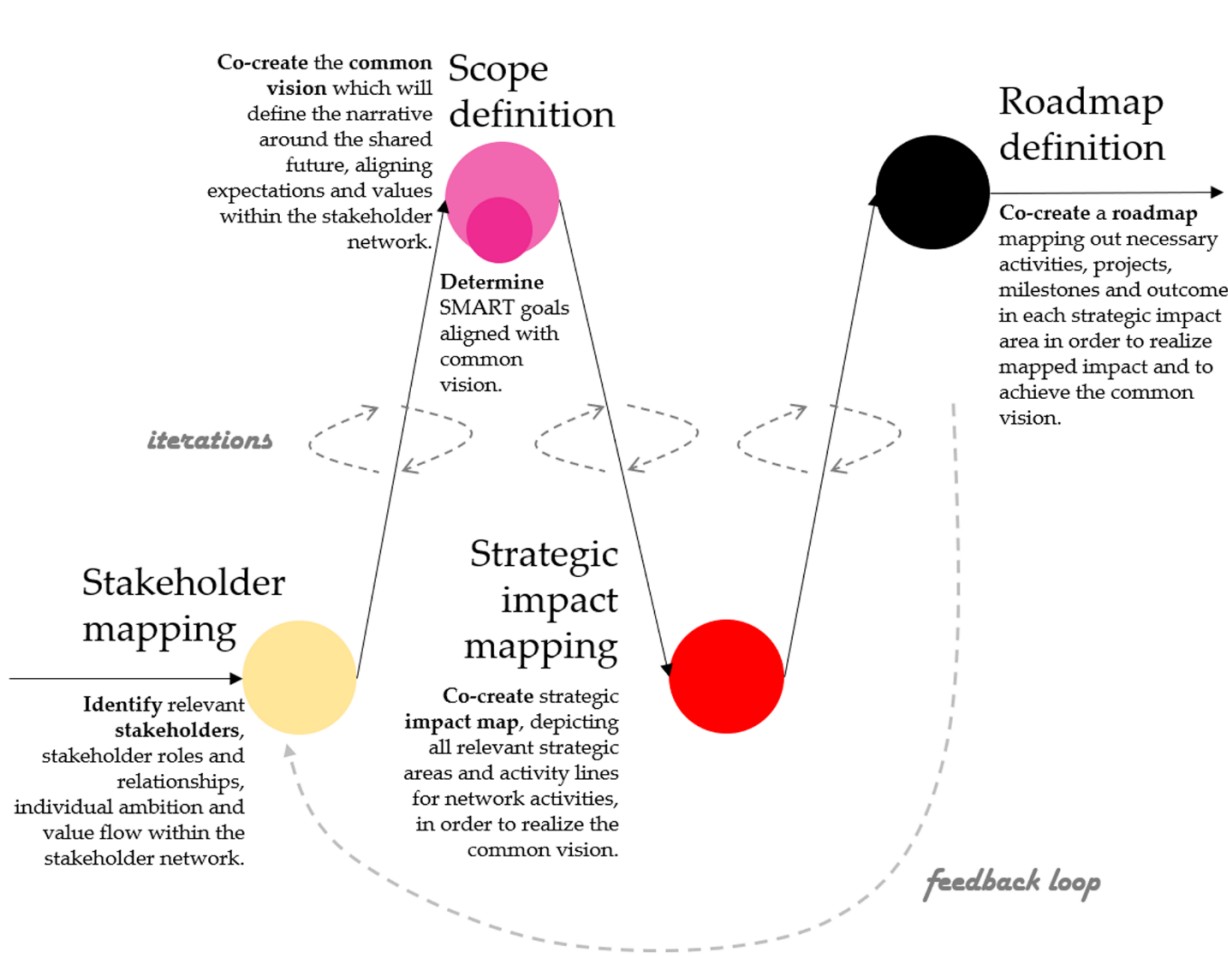

**Figure 2.** Conceptual framework depicting important stages to facilitate collaboration initiation phase.

The proposed framework is built bottom up, is based on co-creation principles, and consists four stages: (1) stakeholder mapping, (2) scope definition, (3) strategic impact mapping, and (4) roadmap definition. In each stage incremental deliverables proposed

in our framework help to transition collaboration from planning towards implementation phase. In the paragraphs below, we describe in more detail each of the stages in our conceptual framework. Additionally, in Table 2, we list required deliverables and suggested co-creation methods to be used in each stage to advance, track, and potentially evaluate the collaboration.

**Table 2.** Methods and deliverables for each collaboration stage.

| | **Stakeholder Mapping** | **Scope Definition** | **Strategic Impact Mapping** | **Roadmap Definition** |
|---|---|---|---|---|
| **Methods** | stakeholder safari, stakeholder interviews, contextual inquiry, survey [60,62] | participatory/co-design session, workshop [63] | participatory/co-design session, generative workshop [63,64] | participatory/co-creation session, generative workshop [63,64] |
| **Deliverables** | stakeholder map (visual), comprehensive description of the stakeholder map, including stakeholder relationships and value flows, overview of individual ambitions, and expectations about the collaboration | description of common vision and list of common goals extracted from common vision | strategic impact map (visual), detailed description of strategic impact map | collaboration roadmap (visual), detailed description of collaboration roadmap, including projects, processes, activities, resources, and temporal considerations, list of targets and target deadlines, review deadlines |

Stakeholder mapping: In the first stage of the proposed framework, collaboration initiation begins with stakeholder mapping. Stakeholder mapping provides an overview of all relevant stakeholders for the collaborative initiative (e.g., living lab)—partners from academia, industry, government, and the non-profit sector, but also potential target and/or user groups are identified. The individual stakeholder profiles, roles, ambitions, goals, and relationships with other stakeholders (e.g., value flows) are then mapped onto the stakeholder map.

Scope definition: In the second stage, the stakeholder network mapped in the previous stage is invited to define the scope of collaboration, by creating a common vision and defining SMART goals for the living lab. Co-creation of a common vision enables alignment of expectations, short- and long-term ambitions, and individual goals within the heterogeneous group of stakeholders. SMART goals allow the stakeholder network to decide strategically on what needs to be done, and when and in what order to realize the common vision.

Strategic impact mapping: In the third stage, the established stakeholder network is invited to generate input for the strategic impact map. The goal of the strategic impact mapping stage is to enable stakeholders to together determine strategy for the living lab. In this stage, stakeholders together evaluate available and necessary capabilities within the living lab stakeholder network, create a comprehensive overview of the strategic areas relevant for scope determined in the previous stage (e.g., obtaining monetary or knowledge resources, bridging society and science) and determine general activity lines-directions—that the network needs to take in order to accomplish common goals (e.g., to obtain missing resources, expand network, establish physical city labs).

Roadmap definition: In the fourth stage, the stakeholder network uses the strategic impact map to determine how previously co-created strategy will be realized. The network is invited to determine required actions, processes, and projects, including milestones and expected outcomes that are crucial to realize the common vision over a set time period. These are then aligned and positioned in chronological order on a roadmap. Stage four bridges the initiation (planning) and operationalization (implementation) phases of the

collaborative initiative. In other words, creating a roadmap is the first step to shifting the collaboration process from planning to implementing intended goals and activities.

## 5. Discussion

While today only through collaborative effort can we successfully face urgent societal challenges, such as climate change or unexpected COVID-19 and the current physical inactivity pandemic, the dynamic nature of collaboration process is highly overlooked in both literature and practice. Thus, the question we raise in the present manuscript is not why to collaborate, but rather, "how to collaborate?" We present a conceptual framework to initiate a discussion towards how to account for the complexity and dynamics of transdisciplinary collaboration in a socio-ecological context, such as living labs. We argue that the collaboration process would significantly benefit from the implementation of a reflective layer, where through an iterative and incremental approach in each stage of collaboration, co-creation principles are employed to support, facilitate, maintain, and evaluate the alignment within a heterogeneous group of stakeholders. In the following paragraphs, we discuss implications and limitations of our work.

Firstly, our conceptual framework proposes to kickstart collaboration with stakeholder mapping—a crucial collaboration stage aimed at understanding stakeholder organization. Selecting the right partners to collaborate with is vital for achieving the project goals [65]. Understanding where these partners come from and how they relate to each other, as well as to the project, is necessary to create and maintain the stakeholder alignment. Stakeholder organization, however, is widely underestimated in practice [66]. Moreover, in our exploratory case study, all three collaborative initiatives focused little to no attention on mapping and understanding the stakeholder network. We, thus, believe that the deliverables proposed in the stakeholder mapping stage of our framework, i.e., stakeholder map, are invaluable tools to understand and monitor stakeholder organization throughout collaboration processes on macro (living lab) and meso (project) levels defined by Schuurman and colleagues [54]. Furthermore, stakeholder mapping is an excellent approach to understanding the balance between common and individual gains as collaboration progresses, especially in collaboration stages focusing on achieving stakeholder alignment through co-creation.

Second, scope definition, strategic impact mapping and the roadmap definition stages in the proposed framework are meant to incrementally create and foster stakeholder alignment by directly involving them in strategy development (scope definition and impact mapping), (project) planning (roadmap definition), and/or revision processes (through the feedback loop). In these stages, we propose to conduct co-creation sessions to together with involved stakeholders co-create deliverables meant to track, monitor, and evaluate the collaboration process and progress. Although more research is needed to further develop and validate comprehensive protocols for these workshops (i.e., co-creation) and deliverables thereof, our framework responds to literature (e.g., [10,30]), which is calling for reliable tools allowing practitioners and academics to track and monitor processes, as well as to measure the impact of collaborative initiatives in living labs and living lab alike settings.

Next, we would like to discuss perhaps the most important contribution of our workhow the proposed conceptual framework could bring more coherence in the living lab operationalization. As co-creation plays an essential role in creating stakeholder alignment, we believe that our framework outlines the opportunities to systematically extend the use of co-creative practices in a living lab context from micro (user/citizen) to meso (project) and macro (living lab) layers defined by Schuurman and colleagues [54]. As we mentioned in the beginning of our manuscript, co-creation in the living lab currently mainly happens on a micro layer [54], meaning it is primarily focused on involving citizens and the end users in research and development processes [11,55]. Considering stakeholder alignment is a major collaboration challenge, co-creation on project (meso) and on living lab (macro) layers may benefit collaboration processes on strategic (living lab) and implementation

(projects in the living lab) levels. Additionally, such an approach provides opportunities to examine and understand how different layers of the living lab ecosystem interrelate; for example, how living lab strategy (e.g., common vision and strategic impact map) is reflected in practice—running projects and other activities (e.g., roadmap). Finally, such an approach could potentially reshape the typical living lab collaborations from project-based, to a more sustainable, strategic, and long-term alliance-like collaboration. This, however, requires further refinement and elaboration of the proposed conceptual framework.

Lastly, we would like to stress that after all, people are the driving force behind any and all collaborative initiatives. Literature suggests [27] that the success of transdisciplinary collaborations strongly depends on interpersonal qualities and traits of collaborating parties, for example, openness, innovative mindset, and willingness to share and embrace transdisciplinary ethics. As was mentioned earlier, with the right tools provided (i.e., our framework), everyone could be a co-creator [47]; however, the effort and personal qualities necessary for preparation and facilitation of the co-creation processes should not be underestimated. In practice, co-creation amongst multiple stakeholders and actors in the living labs does not happen without coordination and support [66] or, as Hirvikoski and Saastamoinen [67] define it, orchestration. Co-creation of collaboration in living labs, as we suggest in our framework, indeed needs to be facilitated. Such an approach, however, demands know-how and, sometimes additional capacity from participating stakeholders [68]. As a result, this calls for holistic and long-term governing mechanisms, such as adaptive governance [68]. Therefore, the next questions to be raised, amongst other things, should concern who is responsible for organizing and facilitating co-creation within multi-stakeholder networks to support transdisciplinary collaboration.

The proposed framework is a first attempt to map the transdisciplinary collaboration processes in living labs. Currently, our framework considers only stages of and tools for the collaboration initiation phase. However, we believe that it provides a strong base for a more extensive framework to be developed in the future, which will consider all phases of transdisciplinary collaboration in living labs and will include an extensive set of guidelines and tools for collaboration in practice. We based the current framework on a narrative literature review and study of our experience in practice. The somewhat lower number of explored empirical cases, the lack of longitudinal approach, and no validation of our conclusions can definitely be considered as limitations of our work. Additionally, in order to account for differences in the unique nature of each collaborative initiatives (e.g., living labs, transdisciplinary academic networks), and to successfully achieve stakeholder alignment and co-create strategy, an extensive amount of work needs to be done in selecting and designing the right tools (i.e., types of workshops proposed in our framework) to successfully facilitate the collaboration process. However, we are well aware that the proposed framework is only in the conceptual phase and needs future validation and refinement. This is, perhaps, the major limitation of our work, which we intend to address in our future practical engagements and research activities.

## 6. Conclusions

There is an evident need for validated frameworks accompanied by practical guidelines on how to set up and implement transdisciplinary collaboration initiatives in complex socio-ecological contexts, such as living labs. In the present manuscript, we propose a conceptual four-stage framework grounded in co-creation. Our main conclusion is twofold. Firstly, our framework illustrates the importance of considering the dynamic nature of collaboration process and, thus, the need for a reflective and iterative approach to transdisciplinary collaboration, both in theory and in practice. Second, we show how co-creation on project (meso) and on living lab (macro) layers not only offers a huge potential to facilitate and maintain so needed stakeholder alignment, but also to better understand the living lab ecosystem as a whole. In particular, how different living lab ecosystem layers (micro, meso, macro) are interrelated.

**Author Contributions:** Conceptualization, I.K. and S.V.; methodology, I.K. and S.V.; formal analysis, I.K.; writing—original draft preparation, I.K.; writing—review and editing, S.V., R.B., Y.L., A.B., T.B., and I.K. All authors have read and agreed to the published version of the manuscript.

**Funding:** This research received no external funding.

**Institutional Review Board Statement:** Not applicable.

**Informed Consent Statement:** Not applicable.

**Data Availability Statement:** Data sharing not applicable.

**Acknowledgments:** This paper is a slightly revised version of an original research article presented in the Top Contributions Session of the Digital Living Lab Days 2020 and published in the conference proceedings.

**Conflicts of Interest:** The authors declare no conflict of interest.

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
