# Peer review of "Facing Societal Challenges in Living Labs: Towards a Conceptual Framework to Facilitate Transdisciplinary Collaborations"

_sustainability, doi:10.3390/su13020614_

Round 1

Reviewer 1 Report

Excellent article on an important subject in the context of Living Labs, particularly on the importance to be given to co-creation with stakeholders. Well supported by the literature review. The cases provide relatively thin support for the conclusions, but are sufficient to support them. The writing style is very good and the practicality of the paper is already very useful.

Author Response

“Excellent article on an important subject in the context of Living Labs, particularly on the importance to be given to co-creation with stakeholders. Well supported by the literature review. The cases provide relatively thin support for the conclusions, but are sufficient to support them. The writing style is very good and the practicality of the paper is already very useful.”

Dear reviewer, thank you for your compliments and positive review of our manuscript. We have completed several changes in the article based on feedback/comments that we received from other reviewers. The changes also include improving the linkage between theoretical and practical observations in Lessons Learned section (now, section 3.2.4). This, hopefully, helps to strengthen our conclusions in the last section of the paper.

Reviewer 2 Report

Review of: “Facing societal challenges in the Living Labs: Towards conceptual framework to facilitate transdisciplinary collaborations”

By: Kalinauskaite Indre, Brankaert Rens, Lu Yuan, Bekker Tilde, Brombacher Aarnout and Steven Vos

__________________________________________________________________________________

Dear authors,

Thank you for the opportunity to review this article. Please, find my comments and suggestions below.

General comments:

Kalinauskaite et al. examine in this article the theoretical and practical challenges of transdisciplinary collaboration in a living lab context. The main research question is how to collaborate? The authors argue that this question has not been sufficiently addressed in the literature on transdisciplinary collaboration and living labs. Based on three case studies the authors have developed a framework that aims to help researchers and practitioners to better design and facilitate complex stakeholder collaborations in living lab settings. The framework focuses on the first phase of the co-creation process, i.e. the collaboration initiation phase, which is characterized by four stages. For each of these four stages, the authors suggest different methods to be used and tasks/deliverables to be executed.

The article is interesting to read and well-structured. The topic is timely, since living labs or variations thereof combined with co-creation approaches are widely used in the European research landscape. I appreciate the focus on the initiation phase of the collaboration, since a successful execution of this stage is crucial for achieving project goals and making impacts in living labs settings. In addition, the subdivision of the collaboration initiation phase into four stages provides an added value compared to the often more general co-creation frameworks. Furthermore, the framework and suggested methods and deliverables are presented in a concise and clear manner. The discussion section provides food for thought and brings together the various elements of the paper nicely. Also discussing the different layers (micro, meso, macro) of the living lab concept is useful and not very common in the living lab literature.

The authors’ objective is to address the gap in literature on how to collaborate in living lab settings. The section that explores this particular body of literature and provides the argument that there is a gap is rather short. Reading sections 2 and 3, I wondered about the many stakeholder engagement handbooks and/or co-creation guidelines produced by (EU-funded) research projects in recent years that are publicly available (see for example: Durham E., Baker H., Smith M., Moore E. & Morgan V., 2014. The BiodivERsA Stakeholder Engagement Handbook) and how these handbooks and guidelines fit with argument that the authors make in the article. I recommended expanding the literature section a bit more in order to address this issue and potentially strengthen the authors’ argument. Expanding the sections 2 and 3 may also help to (partially) overcome the limitations of the authors´research discussed at the end of the article, e.g, the low number of explored cases studies. If expanding the literature sections is not feasible, I would frame the article more strongly as a lessons learnt/best practice paper than as a paper that aims to address the lack in theoretical and practical guidelines on how to collaborate. In addition, it may also benefit the article to highlight already in the introduction the most important contribution of the authors’ work (see lines 323-338) by formulating more specifically the aims and objectives of the article. It will make it easier for the reader to follow along the argument made in the article

Minor comments.

  • Please, add some more recent references on societal challenges in the introduction (e.g. on SDGs).
  • Please, explain the term socio-ecological system. I am familiar with this term in the context of sustainability studies or vulnerability & resilience studies and I am not sure why the term is used in the context of this article. Maybe there are different understanding or uses of this term?
  • Statement made in lines 122-124: how does this fit with handbooks and guidelines produced by other projects (see above)?
  • Figure 2 is blurry and contrast between arrows and dark-colored circles is not optimal. Please, improve.
  • Please, realign paragraph on roadmap definition.
  • Please check the use of the definitive and indefinite articles throughout the text, it seems “the” and “a(n)” are sometimes missing or used when it is not necessary. Some examples:
    • Line 17: In the present paper…
    • Line 30: …to a healthcare crisis caused by a global pandemic
    • Line 36: ….in living lab projects.
    • Line 41: … in the collaboration process.
    • Line 292: …we raise in the present paper
  • Some examples of other editing issues:
    • Lines 30/31: …combat the world’s problems..
    • Lines 269/270: …and when and in what order…
    • Line 290: …face urgent societal …

Author Response

Dear reviewer, thank you for your extensive review of our work and very valuable feedback and suggestions. We attempted to work in most of them. In cases, where we were not able to integrate your feedback, we respond in the text below, providing more extensive explanation. 

The authors’ objective is to address the gap in literature on how to collaborate in living lab settings. The section that explores this particular body of literature and provides the argument that there is a gap is rather short.  <…> Expanding the sections 2 and 3 may also help to (partially) overcome the limitations of the authors´ research discussed at the end of the article, e.g., the low number of explored cases studies. If expanding the literature sections is not feasible, I would frame the article more strongly as a lessons learnt/best practice paper than as a paper that aims to address the lack in theoretical and practical guidelines on how to collaborate.

Thank you for this remark. Indeed, we base our conceptual framework not on the extensive theoretical exploration, but rather on a narrative literature review in combination with the lessons learned in practice. Following your suggestion, we have attempted to clarify our approach in the introduction.

Reading sections 2 and 3, I wondered about the many stakeholder engagement handbooks and/or co-creation guidelines produced by (EU-funded) research projects in recent years that are publicly available (see for example: Durham E., Baker H., Smith M., Moore E. & Morgan V., 2014. The BiodivERsA Stakeholder Engagement Handbook) and how these handbooks and guidelines fit with argument that the authors make in the article. I recommended expanding the literature section a bit more in order to address this issue and potentially strengthen the authors’ argument. <…> Statement made in lines 122-124: how does this fit with handbooks and guidelines produced by other projects (see above)?

On the one hand, every project, living lab, stakeholder and knowledge network is different and will face different challenges depending on how the collaboration process is defined and managed. On the other hand, from scientific perspective various collaborative initiatives, especially in the living lab context share several communalities, such as open innovation context, involvement of citizens, and combination of actors and organizations from different sectors.  Indeed, there are numerous handbooks, produced by successful projects, to support collaborative research and innovation. We also refer to one of these handbooks in section 2.1. What we see today, is that often these handbooks and guides are tied to a specific project and its goals, contexts, or even socio-cultural settings. Although, we may speculate now that they share similarities (e.g., involving co-creation in engaging user), it is evident that we still lack more fundamental frameworks in scientific literature, which would scope the core ingredients and stages of successful collaborations on a theoretical level.

In addition, it may also benefit the article to highlight already in the introduction the most important contribution of the authors’ work (see lines 323-338) by formulating more specifically the aims and objectives of the article. It will make it easier for the reader to follow along the argument made in the article

Thank you for this suggestion, we revised our introduction and improved formulation of our main ambition and objectives.

Please, add some more recent references on societal challenges in the introduction (e.g. on SDGs).

We have revised the first paragraph of introduction and integrated reference to SDGs in the context of societal challenges.

Please, explain the term socio-ecological system. I am familiar with this term in the context of sustainability studies or vulnerability & resilience studies and I am not sure why the term is used in the context of this article. Maybe there are different understanding or uses of this term?

The socio-ecological system in present manuscript, is indeed in line with the definition of these systems in vulnerability and resilience studies, as it refers to real-life urban contexts, considered as holistic systems where the interaction between humans (social) and their environment (ecological) is central (e.g., Young et al., 2006; https://doi.org/10.1016/j.gloenvcha.2006.03.004). This point of view enables inclusion of wide range of individual (e.g., psychological) and environmental (e.g., social as well as physical environment) factors in in-situ research and innovation activities in living labs (e.g., as described by Baccarne et al., 2016; https://timreview.ca/article/972).

Figure 2 is blurry and contrast between arrows and dark-colored circles is not optimal. Please, improve.

We have uploaded higher resolution figure.

Please, realign paragraph on roadmap definition.

We have realigned this paragraph.

Please check the use of the definitive and indefinite articles throughout the text, it seems “the” and “a(n)” are sometimes missing or used when it is not necessary.

Thank you for this comment, we have revised the text and fixed all the mistakes outlined in your examples.

Reviewer 3 Report

The present manuscript on living labs and transdisciplinary collaboration in the Dutch context is both timely and thematically interesting. It clearly falls under the scope of Sustainability. The authors like to "present a conceptual framework to help practitioners and researchers effectively set up and facilitate transdisciplinary collaborations in the complex multi-stakeholder socio-ecological systems, such as living labs". While the theoretical part seems to be well-developed, I particularly recommend to revise the case study part, improving the linkages between theory and practice. Moreover, the theoretical part could be enriched by existing expierences in the field, published in Sustainability or elsewhere. Should the authors be able to carry out a revision, I would strongly recommend its publication.

SECTION 1

- The Introduction is chiefly theoretical in character. To grasp the readers interest at the beginning, I strongly suggest to include existing experiences in the implementation of living labs and transdisciplinary research. For example

https://doi.org/10.3390/su12145593
https://doi.org/10.3390/su12124835
https://doi.org/10.3390/su12031120

- This section could profit from presenting the specific study aim(s) more clearly
- To better link the theoretical and empirical parts, the aims should briefly make clear which case studies are used and why.

SECTIONS 2 and 3

- Sections 2 and 3 could be merged into a "Theoretical background" section.
- In section 3 the authors highlight the need for aligning stakeholders. How can we ensure a high degree of empathy in collaborative planning, to an "alignment" towards the interests of more powerful stakeholders?
- In the current section 3, please clarify the terminological differences between co-creation and co-design or co-production.

SECTION 4

- Here, the authors describe how they selceted the "material," yet they do not tell us much regarding the methods applied (apart from a few words in Table 1). Please clarify how you researched the cases (methods applied, number of people involved/interviewed, etc.); this information could be added to Table 1.
- To make the manuscript visually appealing, photographs from the workshops (or of their outcomes) could be added.

SECTION 5

- I wonder if this section should not be part of a more comprehensive "Results" section, for the concept is, in fact, THE result of this study.

SECTION 6

- I recommend to delete subsection 6.1 (for it is the only subsection) or to include more subsections.

Author Response

Dear reviewer, thank you for your extensive review of our work and very valuable feedback and suggestions. We attempted to work in most of them. In cases, where we were not able to integrate your feedback, we respond in the text below, providing more extensive explanation. 

While the theoretical part seems to be well-developed, I particularly recommend to revise the case study part, improving the linkages between theory and practice.

Thank you for this suggestion. We revised the section learned (now, section 3.2.4) describing lessons in our case studies and improved the linkage between theoretical and practical observations. This, hopefully, helps to strengthen our conclusions in the last section of the paper.

SECTION 1:

Moreover, the theoretical part could be enriched by existing experiences in the field, published in Sustainability or elsewhere. <…> The Introduction is chiefly theoretical in character. To grasp the readers interest at the beginning, I strongly suggest to include existing experiences in the implementation of living labs and transdisciplinary research. For example: https://doi.org/10.3390/su12145593; https://doi.org/10.3390/su12124835; https://doi.org/10.3390/su12031120.

We have revised the corresponding part of the introduction and included additional references to more recent practical examples of how living lab approach is employed to tackle societal challenges.

This section could profit from presenting the specific study aim(s) more clearly. To better link the theoretical and empirical parts, the aims should briefly make clear which case studies are used and why.

Thank you for this suggestion, we revised our introduction and improved formulation of our main ambition and objectives. Additionally, we clarified the last paragraph of the introduction presenting the structure of our manuscript, the modifications intend to provide the reader with a better overview of how we our goals were achieved.

SECTIONS 2 and 3:

Sections 2 and 3 could be merged into a "Theoretical background" section.

We followed your advice and merged sections 2 and 3 into “2. Theoretical background section”.

In section 3 the authors highlight the need for aligning stakeholders. How can we ensure a high degree of empathy in collaborative planning, to an "alignment" towards the interests of more powerful stakeholders?

Co-creation enables dialog between collaborating parties and in turn contributes to transparency in both, individual agendas and reasons for participation of involved stakeholders. From literature and our experience, we see that well-designed co-creation workshops and outcomes thereof can significantly contribute to increasing empathy amongst collaborating parties. We also hint to this in section 3.2.4. of our manuscript. This is likely to happen because co-creation in collaborative setting facilitates a better understanding of our counterpart’s goals, ambitions and needs and subsequent alignment of these factors.

In the current section 3, please clarify the terminological differences between co-creation and co-design or co-production.

We have followed your advice and clarified this difference in the first paragraph of current section 2.2. (lines 130-132).

SECTION 4

Here, the authors describe how they selceted the "material," yet they do not tell us much regarding the methods applied (apart from a few words in Table 1). Please clarify how you researched the cases (methods applied, number of people involved/interviewed, etc.); this information could be added to Table 1.

We have revised the first paragraph of current section 3.1 and explained that cases were analyzed using critical reflection method.

To make the manuscript visually appealing, photographs from the workshops (or of their outcomes) could be added.

We have included a collage of photographs depicting co-creation activities during one of the workshops (specifically, strategic impact mapping workshop described in case B).

SECTION 5

I wonder if this section should not be part of a more comprehensive "Results" section, for the concept is, in fact, THE result of this study.

Indeed, we agree that our framework is the result of our theoretical and practical exploration. However, to not confuse it with a Results the section 3. Collaboration challenges in practice – exploratory case study, we decided to leave the section 4. Proposed conceptual framework as a separate section of the manuscript.

SECTION 6

I recommend to delete subsection 6.1 (for it is the only subsection) or to include more subsections.

We have followed your recommendation to delete the subsection with the limitations heading in the current section 5.

Round 2

Reviewer 3 Report

The revision of the first version of the manuscript reads much better, due to a slightly revised structure, includes more information on methods, supported by photos, and clarifies key terms. The first lines, I feel, now better introduce the reader and the authors better embed their research in literature on practical experiences on living labs. In sum, I think the present paper can really enrich the status quo of living labs research and can be accepted for publication.